# Symptom clusters in chronic kidney disease and their association with people's ability to perform usual activities

Currie Moore[1]*, Shalini Santhakumaran[2], Glen P. Martin[3], Thomas J. Wilkinson[4], Fergus J. Caskey[5,6], Winnie Magadi[2], Rachel Gair[7], Alice C. Smith[4], David Wellsted[1], Sabine N. van der Veer[3]

**1** Health Research Methods Unit, University of Hertfordshire, Hatfield, United Kingdom, **2** UK Renal Registry, Bristol, United Kingdom, **3** Centre for Health Informatics, Division of Informatics, Imaging and Data Sciences, Manchester Academic Health Science Centre, The University of Manchester, Manchester, United Kingdom, **4** Leicester Kidney Lifestyle Team, Department of Health Sciences, University of Leicester, Leicester, United Kingdom, **5** Population Health Sciences, University of Bristol, Bristol, United Kingdom, **6** Renal Unit, North Bristol NHS Trust, Bristol, United Kingdom, **7** Renal Association, Bristol, United Kingdom

* c.moore9@herts.ac.uk

**Data Availability Statement:** The data underlying this article were collected for Transforming Participation in Chronic Kidney Disease (TP-CKD), a national service improvement programme in 14

## Abstract

### Background

People living with a long-term condition, such as chronic kidney disease (CKD), often suffer from multiple symptoms simultaneously, making symptom management challenging. This study aimed to identify symptom clusters in adults with CKD across treatment groups and investigate their association with people's ability to perform their usual activities.

### Methods

We conducted a secondary analysis of both cross-sectional and longitudinal data collected as part of a national service improvement programme in 14 kidney centres in England, UK. This data included symptom severity (17 items, POS-S Renal) and the extent to which people had problems performing their usual activities (single item, EQ-5D-5L). We categorised data by treatment group: haemodialysis (n = 1,462), transplantation (n = 866), peritoneal dialysis (n = 127), or CKD without kidney replacement therapy (CKD non-KRT; n = 684). We used principal component analysis to identify symptom clusters per treatment group, and proportional odds models to assess the association between clusters and usual activities.

### Results

Overall, clusters related to: lack of energy and mobility; gastrointestinal; skin; and mental health. Across groups, the 'lack of energy and mobility' clusters were associated with having problems with usual activities, with odds ratios (OR) ranging between 1.24 (95% confidence interval [CI], 1.21–1.57) for haemodialysis and 1.56 for peritoneal dialysis (95% CI, 1.28–1.90). This association was confirmed longitudinally in haemodialysis (n = 399) and transplant (n = 249) subgroups.

renal centres across England (UK). This data is held by the UK Renal Registry. The authors had no special privileges to access and analyse the data for research purposes. They applied for permission (Ref: UKRR ILD32) through a standard approval process, which is described here: https://renal.org/audit-research/how-access-data/ukrr-data; the data supporting this study may be accessed following this same approval process. Once permission had been received, the data was analysed within the UK Renal Registry environment.

**Funding:** This work was supported by a Kidney Research UK Innovation grant (reference IN_013_20160304; https://kidneyresearchuk.org/research/research-grants). GPM's time was partially supported by funding from the MRC-NIHR Methodology Research Programme (grant number: MR/T025085/1; https://mrc.ukri.org/funding/science-areas/methodology-research). ACS and TJW are grateful for funding support from the Stoneygate Trust. Their contribution is independent research supported by the NIHR Leicester Biomedical Research Centre (https://www.leicesterbrc.nihr.ac.uk). The views expressed in this publication are those of the authors and not necessarily those of the NHS, the NIHR or the Department of Health and Social Care. The funders had no role in the study design, data collection and analysis, decision to publish, or preparation of the manuscript.

**Competing interests:** The authors have declared that no competing interests exist.

## Implications

Our findings suggest that healthcare professionals should consider routinely assessing symptoms in the 'lack of energy & mobility' cluster in all people with CKD, regardless of whether they volunteer this information; not addressing these symptoms is likely to be related to them having problems with performing usual activities. Future studies should explore why symptoms within clusters commonly co-occur and how they interrelate. This will inform the development of cluster-level symptom management interventions with enhanced potential to improve outcomes for people with CKD.

## Introduction

Symptom burden is high in people with long-term conditions [1–4]. This is also true for people with chronic kidney disease (CKD) across disease stages and treatments [5, 6]. Those with CKD may suffer from 6 to 20 symptoms simultaneously [7], which negatively affects their quality of life and increases the risk of treatment non-adherence, health care utilization, and mortality [8–10]. It is therefore not surprising that people with CKD have identified improving symptom management as a research priority [11], and that clinical practice guidelines include it as a key element of CKD care [12, 13]. However, considering the number of concurrent symptoms, it is challenging for healthcare professionals and patients to assess each symptom separately and develop treatment plans accordingly. This may explain why CKD symptom management is often suboptimal [14–16].

One way to address this problem is to develop management strategies that target multiple, potentially related, symptoms at once. This aligns with the suggestion that within and across long-term conditions symptoms often co-occur, i.e. cluster together [17–19]. In CKD, previous research proposed clusters related to fatigue, pain, gastrointestinal, and skin symptoms [20–28]. They also reported negative associations between these clusters and outcomes linked to quality of life, such as health functioning [21, 25–27, 29] and depression [21]. This implies that managing clusters, rather than individual symptoms, may be an effective way to improve outcomes related to quality of life for people with CKD [24] and to reduce the overall challenge faced by both healthcare professionals and people with CKD in considering the number of concurrent symptoms.

CKD symptom clusters have been mainly investigated in people receiving dialysis [28]. Studies considering other kidney replacement therapies (KRTs) or CKD stages often identified clusters across rather than stratified by treatment or stage [20, 21, 23, 26, 27]; one reason for this lack of stratification might have been the modest sample sizes of <450 people [20, 21, 23, 26, 27]. Only few studies looked at symptom clusters specifically for people with a kidney transplant [22] or with CKD but not on KRT (CKD non-KRT) [25], despite these people reporting a similar symptom burden [5, 6, 30]. This leaves it largely unknown to what extent current clusters generalise across or differ between treatment groups.

This study, therefore, aimed to (1) explore symptom clustering in a large data set, stratified by CKD treatment group, (2) assess the relevance of the identified clusters by investigating their association with people's ability to perform their usual activities, and (3) determine if these associations were stable over time. We anticipate the findings to contribute to developing cluster-level symptom management strategies with potential to improve outcomes for people with CKD.

## Methods

We conducted a secondary analyses of cross-sectional and longitudinal data collected in the context of a national service improvement programme in 14 kidney centres across England (UK) called *Transforming Participation in Chronic Kidney Disease* (TP-CKD) [31]. We followed the Strengthening the Reporting of Observational Studies in Epidemiology (STROBE) guidelines [32] (see S1 Table) and the nomenclature for kidney function and disease proposed by Levey and colleagues [33] for reporting our findings.

### Transforming Participation in Chronic Kidney Disease (TP-CKD)

The TP-CKD programme aimed to support people with CKD to better manage and make decisions about their own care and treatment. This included introducing collection of patient-reported outcome data in 14 English kidney centres [34]. Between December 2015 and December 2017, members of local kidney care teams approached eligible patients in their centre. People were eligible if they were aged 18 years or over and receiving care for any stage of CKD or on any form of KRT. The number and sociodemographic characteristics of people on KRT treated in each centre are provided in S2 Table; this does not include people with CKD not on KRT because this data was not available. People who were interested received a paper copy of the questionnaire, which they returned to their unit upon completion. Centres did not record information on whom they screened for eligibility, who had been confirmed eligible but declined participation, or on reasons for declining participation. In twelve centres, people who had previously taken part were invited to complete the questionnaire again at a later time.

### Measures of exposure and outcome

In the current secondary analysis, we used symptom burden as the measure of exposure. To assess exposure, we analysed data collected through the Palliative care Outcome Scale-Symptom (POS-S) Renal [35], which was one of the patient-reported outcome measures included in the paper-based TP-CKD questionnaire. The POS-S Renal consists of 17 symptoms that are common for people with CKD [35], such as pain, weakness or lack of energy, and itching. For each symptom, respondents are asked to indicate to what extent they have been bothered by it over the last week on a scale from 0 (not at all) to 4 (overwhelmingly).

As the outcome measure we used an item from the EuroQol 5 dimensions—5 level (EQ-5D-5L) version [36], which was also included in the TP-CKD questionnaire. The EQ-5D-5L assesses general health status and asks respondents to rate whether they had problems doing their usual activities (e.g. work, study, housework, leisure activities) on a scale from 1 (no problem) to 5 (unable). People with CKD previously reported this item as being important to 'living well' (29). We did not consider the remaining EQ-5D-5L items as part of the outcome measure because: some fully overlapped with symptoms in the POS-S Renal (pain/discomfort; anxiety/depression; mobility); one was not reported by people with CKD as a priority outcome in research (self-care: wash and dress) [37, 38]; and one because its responses could not be scanned into electronic format (self-rated health status).

### Data processing, linkage and ethics

The completed TP-CKD paper-based questionnaires were sent to the UK Renal Registry (UKRR) and scanned into electronic format. The UKKR used people's unique National Health Service (NHS) number to link questionnaires to data on: date of birth; gender; ethnicity; index of multiple deprivation area quintile (proxy of socio-economic status [39] derived from postcode, with higher quintiles representing more social deprivation); treatment group; primary

kidney disease diagnosis; and time on KRT. The UKRR annual report describes the definitions and measurement methods for these data items [40].

Because the primary purpose of TP-CKD was service evaluation and improvement rather than research, no formal ethical approval was required. People taking part in the TP-CKD programme implicitly consented to their data being processed and linked by returning a completed questionnaire(s) to their kidney centre. The UKRR holds permissions under s251 of the NHS Act 2006, to gather, process, and share confidential patient information for the purposes of audit and research. These permissions are renewed annually by the UK's Health Research Authority's Confidentiality Advisory Group. The collection and secondary analysis of the data for this study were approved by the UKRR (ref: UKRR ILD32) and carried out under the ethical permissions granted to the UKRR by the Research Ethics Committee.

### Secondary data analysis

We included data collected during the TP-CKD programme in the secondary analysis if questionnaires had fewer than 4 missing symptom scores for the POS-S Renal and the 'usual activities' item on EQ-5D-5L had been completed. For people who completed a questionnaire on more than one occasion, we used the first questionnaire for the cross-sectional analyses, and the first and last questionnaires for the longitudinal analyses. For the latter, we only included people who were in the same treatment group when completing the first and last questionnaire. In the absence of information on non-responders, we assessed potential selection bias by comparing characteristics of people included in the analysis to those of the overall CKD and KRT population in the UK in 2016, which was the most recent information available at the time.

To identify symptom clusters, we conducted principal component analyses (PCA). We assumed polychoric correlations between symptoms due to the ordinal nature of the scores and stratified by treatment group (CKD non-KRT; peritoneal dialysis; haemodialysis; transplant). We applied oblique promax rotation to account for potential correlation between clusters. For determining the optimal number of clusters we considered: eigenvalues (>1); scree plots; clarity of clustering patterns (i.e. variables loading strongly onto one cluster only; no clusters of single variables); and the variance explained by each cluster [41]. Where there were no clear clustering patterns, we excluded symptoms that were inconsistent with the other symptoms (as indicated by Cronbach's α), and re-ran the PCA [41]. We assigned symptoms to the cluster for which they had the highest factor loading, but only if this loading was >0.5 [27] and was >0.2 higher than loadings for other clusters [20]. We assessed each cluster's internal consistency by calculating Cronbach's α.

For the cross-sectional analyses investigating the association between symptom clusters and people's ability to perform their usual activities, we developed proportional odds models per cluster-treatment group combination. The usual activities score was included as the ordinal outcome, and the total symptom cluster score (i.e. sum of individual symptom scores within the cluster) as a continuous predictor. In addition to unadjusted models (i.e. only including the cluster score), we also developed partially adjusted (i.e. also including age, gender, ethnicity, socio-economic status, and time on KRT) and fully adjusted models (i.e. additionally including scores of the other clusters and of symptoms not assigned to any cluster). We checked for violations of the proportional odds assumption and for non-linearity of covariate effects.

To handle missing data on symptoms, ethnicity, gender and socio-economic status, we used multiple imputation with fully conditional specification, assuming the data were missing at random. We carried out 20 imputations using all available symptom scores, usual activity scores, clinical and socio-demographic data as predictor variables. We checked distributions and correlations between variables, comparing imputed and observed data. Since the UKRR only has full coverage of people currently on KRT, we classified questionnaires as pertaining to

the 'CKD non-KRT' group if they could not be linked to a UKRR record and the type of KRT was missing in the questionnaire.

To confirm the findings from the cross-sectional analyses, we conducted longitudinal analyses using the first and last questionnaire from the subsample of people who had been followed up over time. We developed the same set of proportional odds models using the clusters identified in the cross-sectional analyses, but now with (a) a *within-person change* in usual activities score as an ordinal outcome (decreased; stayed the same; increased) and a *within-person change* in total symptom cluster score as a continuous predictor; and (b) including baseline usual activities and cluster scores and time between surveys as additional covariates in the partially and fully adjusted models.

For all analyses, we used SAS version 9.4 and considered a *p* value of <0.5 significant. We did not perform any sensitivity analyses.

## Results

### Characteristics of people in the TP-CKD programme included in the analyses

The UKRR received questionnaires from 3,421 people, of whom 282 were excluded because they had ≥4 missing symptom scores or did not complete the 'usual activities' item on EQ-5D-5L (see S3 Table for their characteristics). The remaining 3,139 people who completed at least one TP-CKD questionnaire were included in the analysis. Table 1 shows their baseline characteristics. Overall, demographic characteristics of those included in our analyses were similar to characteristics of the overall CKD and KRT population in the UK (see S4 Table). Across the sample, people reported a median of five symptoms (interquartile range (IQR), 3 to 8). Weakness, poor mobility, and difficulty sleeping were the most prevalent symptoms (see Fig 1). Across the total sample, 22% of people reported having severe problems performing their usual activities.

A subsample of 699 people completed follow-up questionnaires at a median of 203 days (IQR, 133 to 301) after baseline, while remaining in the same treatment group. This subsample were mostly people on haemodialysis (n = 399) and with a transplant (n = 249). Their baseline characteristics were comparable to those of the overall sample (see S5 Table).

### Symptom clusters by treatment group

Fig 2 shows the clusters we identified and the symptoms within them.

Table 2 displays in more detail the final assignment of symptoms to clusters, their factor loadings, and each clusters' Cronbach's α (see S6 Table for factor loadings prior to final cluster composition).

Based on the key symptoms making up the clusters, we labelled clusters as being related to: lack of energy and mobility; gastrointestinal; mental health; and skin. Post-hoc analyses to explore why pain was in the 'lack of energy and mobility' (CKD non-KRT, haemodialysis, and transplant) and diarrhoea in the 'mental health' cluster (peritoneal dialysis) suggested that this was because they strongly correlated with the most prominent symptom in that cluster (i.e. poor mobility and feeling anxious, respectively).

### Relationship between symptom clusters and people's ability to do usual activities

Table 3 displays the unadjusted, partially, and fully adjusted proportional odds ratios for the cross-sectional associations between people's total symptom cluster scores and their ability to do usual activities. There was an indication of violation of the proportional odds assumption

**Table 1. Baseline characteristics of people included in our analyses (values are numbers (% after excluding missing), unless indicated otherwise).**

| | *All* | *CKD non-KRT [a]* | *Peritoneal dialysis* | *Haemodialysis* | *Transplant* |
|---|---|---|---|---|---|
| Total n (first survey) | 3139 (100) | 684 (21.8) | 127 (4.1) | 1462 (46.6) | 866 (27.6) |
| Gender (male) | 1636 (61.8) | 147 (62.8) | 73 (61.9) | 881 (61.4) | 535 (62.0) |
| Missing | 490 | 450 | 9 | 28 | 3 |
| Age (Mean, SD) | 61.4, 16.0 | 63.3, 17.1 | 62.4, 16.0 | 64.5, 15.2 | 54.4, 14.1 |
| Missing | 2 | 2 | 0 | 0 | 0 |
| Ethnicity | | | | | |
| White | 2170 (81.0) | 239 (89.9) | 96 (81.4) | 1052 (73.5) | 783 (90.8) |
| Asian | 301 (11.2) | 21 (7.9) | 10 (8.5) | 224 (15.6) | 46 (5.3) |
| Black | 159 (5.9) | 4 (1.5) | 9 (7.6) | 128 (8.9) | 18 (2.1) |
| Other | 48 (1.8) | 2 (0.8) | 3 (2.5) | 28 (2.0) | 15 (1.7) |
| Missing | 461 | 418 | 9 | 30 | 4 |
| Social deprivation [b] | | | | | |
| IMD Quintile 1 (least deprived) | 485 (15.6) | 105 (15.8) | 28 (22.1) | 186 (12.7) | 166 (19.2) |
| IMD Quintile 2 | 529 (17.0) | 137 (20.6) | 29 (22.8) | 191 (13.1) | 172 (19.9) |
| IMD Quintile 3 | 514 (16.5) | 113 (17.0) | 22 (17.3) | 236 (16.2) | 143 (16.5) |
| IMD Quintile 4 | 656 (21.0) | 127 (19.1) | 20 (15.8) | 303 (20.8) | 206 (23.8) |
| IMD Quintile 5 (most deprived) | 934 (30.0) | 183 (27.5) | 28 (22.1) | 544 (37.3) | 179 (20.7) |
| Missing | 21 | 19 | 0 | 2 | 0 |
| Time on KRT in years (Mean, SD) | 7.9, 8.6 | N/A | 2.3, 3.5 | 5.3, 6.9 | 12.9, 9.2 |
| Primary kidney diagnosis | | | | | |
| Diabetes | 501 (19.7) | 31 (23.0) | 27 (23.1) | 359 (25.1) | 84 (9.7) |
| Glomerulonephritis | 467 (18.3) | 27 (20.0) | 19 (16.2) | 215 (15.0) | 206 (23.9) |
| Hypertension | 190 (7.5) | 6 (4.4) | 14 (12.0) | 131 (9.2) | 39 (4.5) |
| Polycystic kidney disease | 257 (10.1) | 14 (10.4) | 10 (8.5) | 83 (5.8) | 150 (17.4) |
| Pyelonephritis | 246 (9.7) | 6 (4.4) | 11 (9.4) | 134 (9.4) | 95 (11.0) |
| Kidney vascular disease | 102 (4.0) | 10 (7.4) | 7 (6.0) | 70 (4.9) | 15 (1.7) |
| Other | 384 (15.1) | 17 (12.6) | 14 (12.0) | 195 (13.6) | 158 (18.3) |
| Uncertain aetiology | 398 (15.6) | 24 (17.8) | 15 (12.8) | 243 (17.0) | 116 (13.4) |
| Missing | 594 | 549 | 10 | 32 | 3 |
| Where questionnaire was completed | | | | | |
| At home | 688 (22.6) | 51 (7.7) | 53 (42.1) | 454 (32.0) | 130 (15.4) |
| In clinical settings (outpatient clinic, kidney unit, GP practice) | 2358 (77.4) | 609 (92.3) | 73 (57.9) | 963 (67.9) | 713 (84.6) |
| Missing | 93 | 24 | 1 | 45 | 23 |
| How questionnaire was completed | | | | | |
| Alone, without support | 1910 (63.4) | 432 (66.5) | 81 (65.3) | 708 (50.4) | 689 (82.7) |
| With support from family member/friend | 595 (19.8) | 178 (27.4) | 29 (23.4) | 282 (20.1) | 106 (12.7) |
| With support from clinical staff | 506 (16.8) | 40 (6.2) | 14 (11.3) | 414 (29.5) | 38 (4.6) |
| Missing | 128 | 34 | 3 | 58 | 33 |
| **Exposure & outcome** [c] | | | | | |
| *POS-S Renal* | | | | | |
| Number of symptoms (median, IQR) [d] | 5 (3,8) | 5 (3,8) | 5 (3,7) | 6 (3,9) | 4 (2,7) |
| *EQ-5D-5L* | | | | | |
| Problems with usual activities [e] | | | | | |
| No problems | 1034 (32.9) | 236 (34.5) | 44 (34.7) | 325 (22.2) | 429 (49.5) |
| Slight problems | 695 (22.1) | 160 (23.4) | 37 (29.1) | 322 (22.0) | 176 (20.3) |
| Moderate problems | 726 (23.1) | 170 (24.9) | 23 (18.1) | 368 (25.2) | 165 (19.1) |
| Severe problems | 434 (13.8) | 76 (11.1) | 18 (14.2) | 264 (18.1) | 76 (8.8) |

*(Continued)*

**Table 1.** (*Continued*)

| | *All* | *CKD non-KRT* [a)] | *Peritoneal dialysis* | *Haemodialysis* | *Transplant* |
|---|---|---|---|---|---|
| Unable to do usual activities | 250 (8.0) | 42 (6.1) | 5 (3.9) | 183 (12.5) | 20 (2.3) |
| Any problems related to [f)] | | | | | |
| Mobility | 2130 (68.3) | 436 (64.0) | 89 (70.6) | 1166 (80.3) | 439 (51.0) |
| Self-care | 1140 (36.5) | 198 (29.0) | 38 (30.2) | 707 (48.6) | 197 (22.9) |
| Pain/discomfort | 1955 (62.6) | 417 (61.6) | 75 (59.1) | 1003 (68.8) | 460 (53.4) |
| Anxiety/depression | 1583 (50.7) | 354 (52.2) | 64 (50.4) | 795 (54.8) | 370 (42.8) |

Note. CKD non-KRT, people with chronic kidney disease not receiving kidney replacement therapy; EQ-5D-5L, EuroQOL Five Dimensions—5 levels version; GP, general practitioner; IMD, index of multiple deprivation; IQR, interquartile range; N/A, not applicable; POS-S Renal: Palliative care Outcome Scale-Symptom Renal; KRT: kidney replacement therapy; SD: standard deviation.

[a)] This included people with any stage of CKD not on KRT who were under the treatment of a kidney centre. The dataset did not contain information on the stage of CKD at enrollment.

[b)] Based on index of multiple deprivation quintiles (27)

[c)] For people who completed more than one questionnaire, we used the response from their first questionnaire for this table

[d)] Refers to number of symptoms with a score of >1 (i.e. reports of being at least slightly bothered by a symptom) from the POS-S Renal

[e)] Domain from the EQ-5D-5L; primary outcome measure of the current study

[f)] Refers to the remaining four dimensions within the EQ-5D-5L where people scored >1 (i.e. reports of having at least slight problems)

in some models where comparing a score of 5 versus a score of 4 and lower, but because few people reported a score of 5 (i.e., unable to do my usual activities) this only affected a small proportion of the data.

Across all clusters and treatment groups, the unadjusted and partially adjusted models showed a significant association between total cluster symptom scores and having problems with usual activities. However, after fully adjusting for other clusters and individual symptoms, only the association between the 'lack of energy and mobility' cluster and problems with usual activities remained significant across treatment groups (i.e. accounting for the 'lack of energy and mobility' cluster attenuated the ORs for other symptom clusters). The fully adjusted models showed that haemodialysis patients had 24% (95% confidence interval (CI), 21–27%) higher odds of having more problems with doing their usual activities for each unit increase in the total score for the 'lack of energy and mobility' cluster. This was 32% (95% CI, 26–38%), 56% (95% CI, 28–90%) and 37% (95% CI, 32–42%) for the CKD non-KRT, peritoneal dialysis and transplant groups, respectively. Other associations remaining significant after full adjustment were those for the 'mental health' cluster for the CKD non-KRT group (OR, 1.21; 95% CI, 1.1–1.33) and the 'skin' cluster for haemodialysis (OR, 1.03; 95% CI, 1.00–1.07) and transplant groups (OR, 0.86; 95% CI, 0.77–0.95).

Since only 6% (n = 43) and 1% (n = 8) of CKD non-KRT and peritoneal dialysis of people in the these groups completed follow-up questionnaires, we limited the longitudinal analyses to haemodialysis and transplant groups (see S7 Table for a summary of changes in total symptom cluster scores and changes in problems with usual activities per treatment group). We found that within-person changes in people's 'lack of energy and mobility' cluster score were associated with higher odds of having increased problems with performing usual activities. This confirms the findings from the cross-sectional analyses, except for the 'skin' cluster (see Table 4).

## Discussion

### Summary of the findings

This study found that, overall, CKD symptom clusters related to lack of energy and mobility, gastrointestinal, skin, and mental health. Although clusters varied between treatment groups,

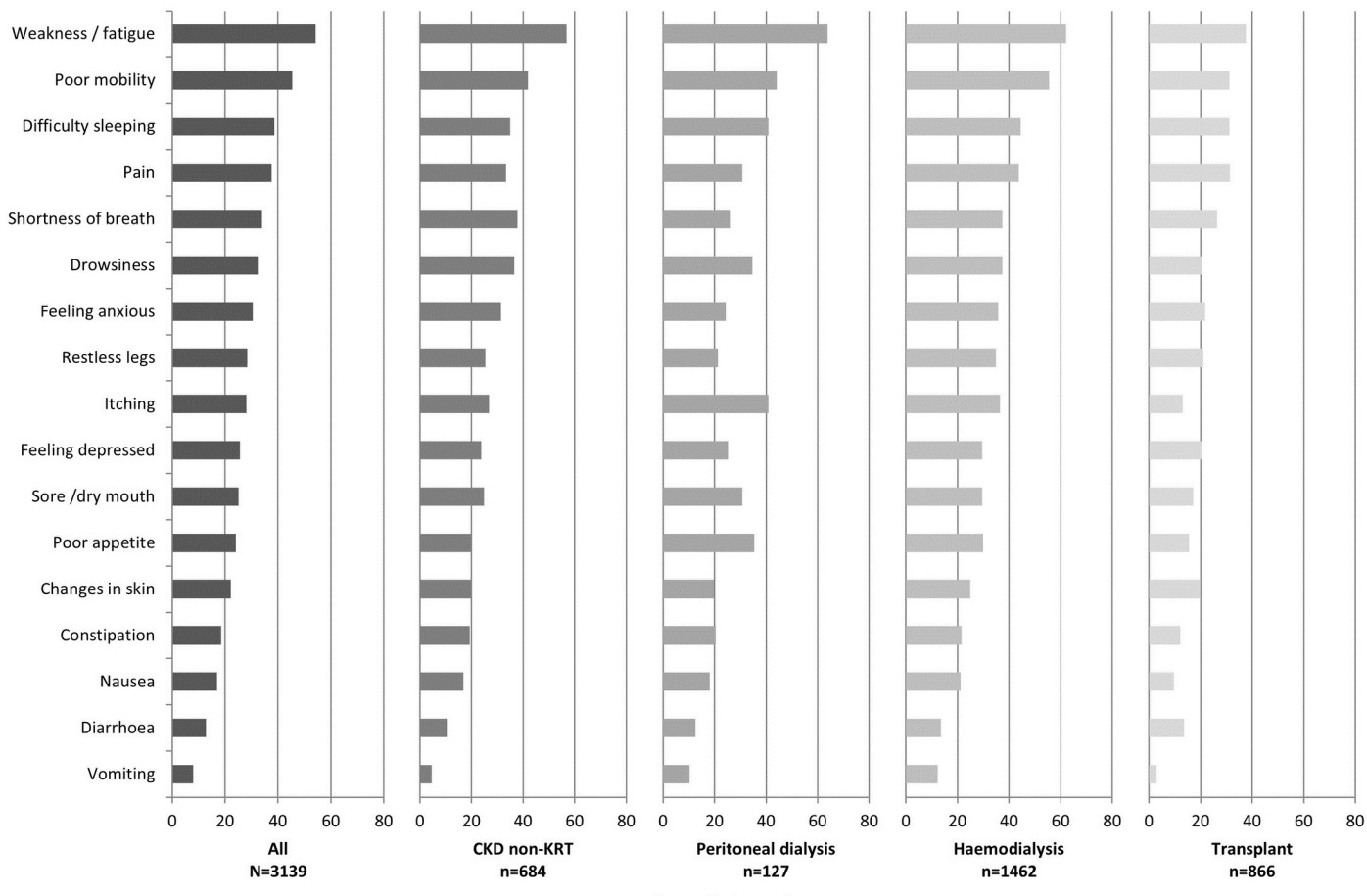

**Fig 1. Prevalence of individual symptoms, presented by treatment group; people were considered to have the symptom if they had a score of > 1 on the POS-S Renal (i.e. reported being at least slightly bothered by a symptom).**

the 'lack of energy and mobility' clusters were consistently associated with having problems doing usual activities across groups, both cross-sectionally and longitudinally.

## Relation to other studies

We used the largest data set to date to investigate symptom clusters for people with CKD. It provides a robust external validation of clusters across treatment groups identified by previous studies, including skin [21, 25–27], gastrointestinal [20, 23], and mental health [25, 26], which suggests that our findings are generalisable beyond the study context. The alignment with previous research on CKD symptom clusters is in contrast with other disease areas, such as cancer, where there seems to be less consistency in symptom clusters across studies [19].

In keeping with previous research [21, 23, 25–27], we found that lack of energy, problems with mobility, pain, mental health, and shortness of breath were the most frequently co-occurring symptoms. Whereas we labelled this cluster as 'lack of energy and mobility' across all treatment groups, other studies referred to it as 'uraemic'. Another difference was that in other studies, it included dizziness [21, 27] and difficulty concentrating [20, 26], which were not listed in the POS-S Renal, the measure of symptom burden in our study. Furthermore, the 'uraemic' cluster in other studies included nausea and poor appetite [20, 21, 26, 27], whereas in our study,

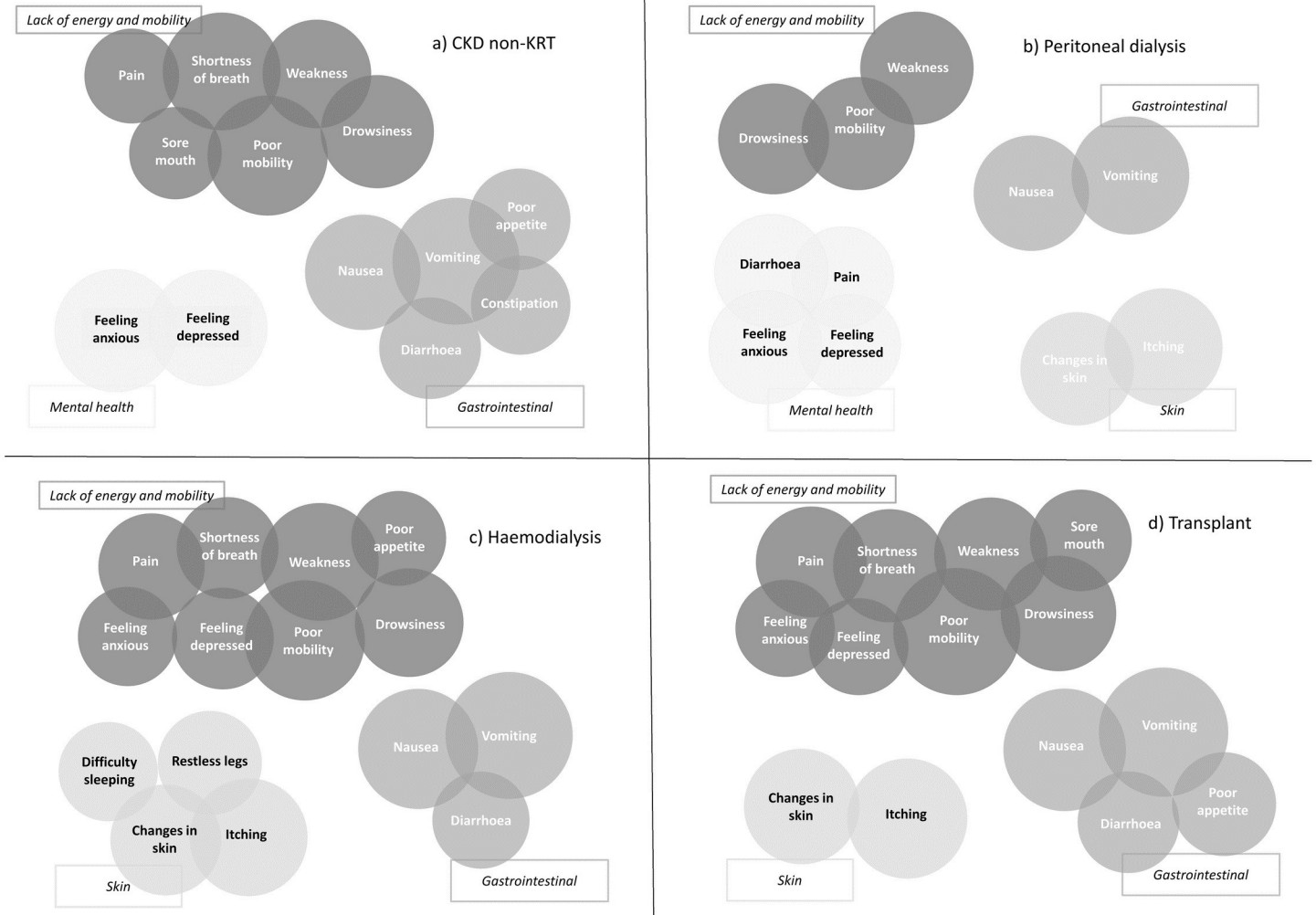

**Fig 2.** Symptom clusters from principal component analyses (PCA), stratified by treatment group; (a) CKD non-KRT, (b) peritoneal dialysis, (c) haemodialysis and (d) transplant. The size of the circles is proportional to the PCA loading which indicates how prominent the variable is within that cluster.

these symptoms, together with vomiting, formed a gastrointestinal cluster. This finding strongly aligns with wider symptom cluster research commonly identifying a separate gastrointestinal cluster consisting of nausea and vomiting [19]. Lastly, unlike other studies [20, 21, 27], we did not find neuro-muscular clusters because the type of symptoms within these (e.g. muscle soreness, numbness or tingling of feet) were not part of our symptom burden score.

The POS-S Renal also does not include fatigue, which is a common symptom in people with CKD that others found to pertain to several symptom clusters [20]. Lack of energy and weakness represent some but not all aspects of fatigue, which is a multi-faceted symptom related to sociodemographic, psychological, clinical and biochemical factors [42]. This implies that part of the people reporting lack of energy and weakness in our study might have actually suffered from fatigue, which may partly explain the strong associations we found between 'lack of energy and weakness' clusters and people's ability to do usual activities.

In the CKD non-KRT treatment group, we found that frequently co-occurring mental health symptoms, i.e. feeling anxious and feeling depressed, formed a stand-alone cluster, which differs from what Lee and Jeon reported in their smaller, Korean study [25]. An

**Table 2. Clusters, symptom loadings and internal consistency (Cronbach's α) resulting from principal component analyses.**

| Symptom clusters | CKD non-KRT | | | Peritoneal dialysis | | | | Haemodialysis | | | Transplant | | |
|---|---|---|---|---|---|---|---|---|---|---|---|---|---|
| | Lack of energy & mobility | GI | Mental health | Lack of energy & mobility | GI | Skin | Mental health | Lack of energy & mobility | GI | Skin | Lack of energy & mobility | GI | Skin |
| *Individual symptoms* | | | | | | | | | | | | | |
| Pain | 0.52 | | | | | | 0.62 | 0.66 | | | 0.72 | | |
| Shortness of breath | 0.82 | | | | | | | 0.61 | | | 0.75 | | |
| Weakness | 0.69 | | | 0.75 | | | | 0.82 | | | 0.74 | | |
| Nausea | | 0.80 | | | 0.78 | | | | 0.85 | | | 0.88 | |
| Vomiting | | 0.95 | | | 0.83 | | | | 0.94 | | | 0.98 | |
| Poor appetite | | 0.61 | | | | | | | 0.53 | | | 0.65 | |
| Constipation | | 0.58 | | - | - | - | - | | | | | | |
| Sore/dry mouth | 0.51 | | | | | | | | | | 0.62 | | |
| Drowsiness | 0.74 | | | 0.73 | | | | 0.70 | | | 0.79 | | |
| Poor mobility | 0.84 | | | 0.76 | | | | 0.86 | | | 0.96 | | |
| Itching | | | | | | 0.83 | | | | 0.83 | | | 0.84 |
| Difficulty sleeping | | | | | | | | | | 0.59 | | | |
| Restless legs | | | | - | - | - | - | | | 0.63 | | | |
| Changes in skin | | | | | | 0.74 | | | | 0.71 | | | 0.78 |
| Diarrhoea | | 0.61 | | | | | 0.75 | | 0.55 | | | 0.61 | |
| Feeling anxious | | | 0.89 | | | | 0.76 | 0.59 | | | 0.55 | | |
| Feeling depressed | | | 0.79 | | | | 0.61 | 0.60 | | | 0.55 | | |
| *Cronbach's α* | 0.82 | 0.72 | 0.76 | 0.74 | 0.85 | 0.56 | 0.74 | 0.85 | 0.73 | 0.71 | 0.88 | 0.72 | 0.54 |
| *Variance* (%) | 44.6 | 7.5 | 6.3 | 40.3 | 9.2 | 7.2 | 10.7 | 43.9 | 5.9 | 7.0 | 49.3 | 7.1 | 6.8 |

Note. GI, gastrointestinal; CKD non-KRT, people with chronic kidney disease not receiving kidney replacement therapy

explanation for mental health being a separate cluster may be that, as people approach kidney failure, they often experience more psychological symptoms [7]. In the peritoneal dialysis group in the current study, this cluster also included pain and diarrhoea, both of which may negatively affect mental health [43, 44].

Our analyses showed that the 'lack of energy and mobility' clusters were associated with problems performing usual activities in all groups and models, which echoes the findings of previous research in which uraemic clusters were associated with impaired physical and mental functioning [20, 26, 27, 29]. For the haemodialysis and transplant groups, this association remained significant over time, similar to what Ng *et al.* found [26]. Additionally, an increase in the CKD non-KRT group's mental health cluster score was associated with increased odds of problems with usual activities, which complements the findings of Lee and Jeon [25]. Compared to our findings, some previous research in people with CKD found more clusters to be associated with patient-reported outcomes [28]. This may be partly explained by the fact that the exposure and outcome measures in these studies often overlapped, thereby introducing an element of circular reasoning in their design. For example, 'pain' was measured both as a symptom (exposure) and as an aspect of physical functioning (outcome) in several studies [21, 24–27]. We addressed this by using a single item to measure the outcome, which reflected the patient-important outcome of 'living well' [37], thus reducing overlap with symptoms in the POS-S Renal.

**Table 3. Association between total symptom cluster scores and ability to perform usual activities.**

| Symptom clusters | CKD non-KRT | | | Peritoneal dialysis | | | Haemodialysis | | | Transplant | | |
|---|---|---|---|---|---|---|---|---|---|---|---|---|
| | Unadjusted | Partially adjusted | Fully adjusted | Unadjusted | Partially adjusted | Fully adjusted | Unadjusted | Partially adjusted | Fully adjusted | Unadjusted | Partially adjusted | Fully adjusted |
| **Lack of energy & mobility** | 1.40 (1.35,1.46) | 1.40 (1.35,1.46) | 1.32 (1.26,1.38) | 1.61 (1.4,1.86) | 1.76 (1.5,2.07) | 1.56 (1.28,1.9) | 1.24 (1.22,1.26) | 1.25 (1.23,1.27) | 1.24 (1.21,1.27) | 1.33 (1.3,1.37) | 1.35 (1.31,1.39) | 1.37 (1.32,1.42) |
| **GI** | 1.30 (1.24,1.37) | 1.34 (1.27,1.41) | 1.02 (0.96,1.08) | 1.58 (1.3,1.93) | 1.29 (1.08,1.55) | 1.10 (0.84,1.43) | 1.30 (1.25,1.36) | 1.33 (1.27,1.39) | 1.00 (0.95,1.05) | 1.39 (1.31,1.47) | 1.41 (1.33,1.5) | 0.96 (0.9,1.04) |
| **Skin** | | | | 1.29 (1.08,1.53) | 1.64 (1.32,2.04) | 1.15 (0.92,1.45) | 1.24 (1.21,1.28) | 1.26 (1.23,1.3) | 1.03 (1.00,1.07) | 1.33 (1.23,1.44) | 1.30 (1.2,1.41) | 0.86 (0.77,0.95) |
| **Mental health** | 1.56 (1.45,1.69) | 1.64 (1.51,1.78) | 1.21 (1.1,1.33) | 1.30 (1.16,1.44) | 1.37 (1.21,1.54) | 1.10 (0.95,1.27) | | | | | | |

**Note.** GI, gastrointestinal; CKD non-KRT, people with chronic kidney disease not receiving kidney replacement therapy; Values are odds ratios (95% confidence intervals), reflecting the odds of having more[a] problems with performing usual activities per unit increase in total symptom cluster score; Partially adjusted—adjusted for age, sex, ethnicity, Index of Multiple Deprivation (IMD) quintile and time on kidney replacement therapy (KRT, for haemodialysis, peritoneal dialysis and transplant groups only). Fully adjusted–adjusted for all variables in the partially adjusted model, and additionally for the total scores of all other symptom clusters and scores of any individual symptoms not appearing in any cluster. Different symptoms comprised each cluster per group: *Lack of energy and mobility* (CKD non-KRT– 6 symptoms—pain, shortness of breath, weakness, sore/dry mouth, drowsiness, poor mobility; Peritoneal dialysis– 3 symptoms–weakness, drowsiness, poor mobility; Haemodialysis– 8 symptoms—pain, shortness of breath, weakness, poor appetite, drowsiness, poor mobility, feeling anxious, feeling depressed; Transplant– 8 symptoms—pain, shortness of breath, weakness, sore/dry mouth, drowsiness, poor mobility, feeling anxious, feeling depressed); *GI* (CKD non-KRT– 5 symptoms–nausea, vomiting, poor appetite, constipation, diarrhoea; Peritoneal dialysis– 2 symptoms–nausea and vomiting; Haemodialysis– 3 symptoms–nausea, vomiting, diarrhoea; Transplant– 4 symptoms–nausea, vomiting, poor appetite, diarrhoea), *Skin* (CKD non-KRT–not applicable; Peritoneal dialysis– 2 symptoms–itching, changes in skin; Haemodialysis– 4 symptoms–itching, difficulty sleeping, restless legs, changes in skin; Transplant—2 symptoms–itching, changes in skin), *Mental health* (CKD non-KRT– 2 symptoms—feeling anxious, feeling depressed; Peritoneal dialysis– 4 symptoms–pain, diarrhoea, feeling anxious, feeling depressed).

[a] 'More' refers to any one step increase on the ordinal scale of the outcome measure (e.g. from no to slight problems, or from severe to extreme problems)

## Limitations

One limitation of our study is the lack of information on which people were selected and invited to complete the TP-CKD questionnaire, and what the characteristics were of those who declined. Although people included in our analyses were representative for the overall CKD population in the UK, we cannot rule out the potential presence of selection bias in our data set. In addition, the data analysed in this study was collected in 2017, and we propose repeating the analysis in the future when updated registry data becomes available.

Furthermore, we did not have access to comorbidity data. Comorbidities are common in people with CKD [45] and may cause symptoms that were not part of our symptom burden score. Furthermore, comorbidities may have a negative impact on people's ability to perform daily activities by increasing the chance of health-related problems, such as medication side effects and hospital admissions. Comorbidities are, therefore, an unmeasured confounder in our analyses that might have explained part of the variation in our outcome measure.

Lastly, due to the small number of people in the CKD non-KRT and peritoneal dialysis groups who completed the questionnaire on more than one occasion, we could not perform a longitudinal analysis to confirm the findings from the cross-sectional analysis in these groups. Furthermore, variation in the timing of follow-ups did not allow assessment of the stability of the symptom clusters over time. The stability of symptom clusters over time has direct implications on symptom management strategies and is an important next step in symptom cluster research [17, 19, 28].

**Table 4. Association between within-person change in total symptom cluster scores and within-person change in ability to perform usual activities.**

| Symptom clusters | Haemodialysis | | | Transplant | | |
|---|---|---|---|---|---|---|
| | Unadjusted | Partially Adjusted | Fully adjusted | Unadjusted | Partially Adjusted | Fully adjusted |
| Lack of energy & mobility | 1.13 (1.09,1.18) | 1.22 (1.16,1.29) | 1.23 (1.16,1.31) | 1.29 (1.19,1.39) | 1.53 (1.37,1.71) | 1.50 (1.32,1.71) |
| Gastrointestinal | 1.10 (1.00,1.21) | 1.21 (1.07,1.37) | 0.95 (0.82,1.1) | 1.30 (1.12,1.52) | 1.46 (1.22,1.76) | 0.95 (0.75,1.2) |
| Skin | 1.12 (1.05,1.19) | 1.15 (1.06,1.24) | 0.97 (0.89,1.07) | 1.13 (0.95,1.35) | 1.41 (1.11,1.8) | 1.11 (0.81,1.53) |

**Note.** Values are odds ratios (95% confidence intervals), reflecting the odds of having increased problems with performing usual activities per unit increase in the change in total symptom cluster score; Partially adjusted—adjusted for age, sex, ethnicity, Index of Multiple Deprivation (IMD) quintile and time on kidney replacement therapy. Fully adjusted–adjusted for all variables in the partially adjusted model, and additionally for the total scores of all other symptom clusters and scores of any individual symptoms not appearing in any cluster. Different symptoms comprised each cluster per group (underline denotes symptoms appearing in one patient group but not the other): *Lack of energy and mobility* (8 symptoms in each, Haemodialysis—pain, shortness of breath, weakness, poor appetite, drowsiness, poor mobility, feeling anxious, feeling depressed; Transplant—pain, shortness of breath, weakness, sore/dry mouth, drowsiness, poor mobility, feeling anxious, feeling depressed); *Gastrointestinal* (Haemodialysis– 3 symptoms–nausea, vomiting, diarrhoea; Transplant– 4 symptoms–nausea, vomiting, poor appetite, diarrhoea), *Skin* (Haemodialysis– 4 symptoms–itching, difficulty sleeping, restless legs, changes in skin; Transplant—2 symptoms–itching, changes in skin).

## How healthcare professionals could use the study findings

The study findings provide pointers for how healthcare professionals could improve CKD symptom assessments in clinic to reduce symptom underreporting and undertreatment. Although the need to assess and treat people's symptom burden has been recognised [46], many symptoms remain unreported [47] and go untreated despite the availability of nonpharmacological and pharmacological interventions [16, 48]. Current guidelines recommend the assessment of symptoms by validated measures [13], such as the POS-S Renal [46]. However, these assessment tools are often long lists of individual symptoms, and time-constrained clinics are unlikely to be able to assess each symptom separately.

Based on the study findings, healthcare professionals could consider the following stepwise, more efficient cluster-based approach to symptom assessment:

1. **Ask people with CKD an open-ended question about what symptoms are bothering them**; even though our analyses indicate which co-occurring symptoms are related to problems with usual activities across treatment group, they do not necessarily reflect which symptoms matter at the individual level. Therefore, opening the discussion with this question remains the most straightforward and personalised way of assessing symptom burden.

2. **If a person volunteers a symptom, pro-actively ask about the other symptoms in the cluster to which that symptom pertains** because our analyses showed that they frequently co-occur. People do not always volunteer all symptoms at clinic appointments for a variety of reasons, such as not knowing that some symptoms may be related to their CKD, or assuming there is no treatment available [15, 49]. Encouraging people with CKD to report specific symptoms could address this, especially since some may withhold this information until prompted by a healthcare professional [49].

3. **If a person does not volunteer any symptoms, ask specifically about symptoms in the 'lack of energy and mobility' cluster.** In all treatment groups, these were the most commonly reported and consistently associated with problems with usual activities. It is important to note is that mental health symptoms are key components of this cluster for the haemodialysis and transplant groups and formed a unique cluster for the CKD non-KRT treatment group, also with significant associations with problems with usual activities.

In addition, the results from our longitudinal analyses in the haemodialysis and transplant groups suggested that healthcare professionals might consider monitoring within-patient increases in the 'lack of energy and mobility' cluster score because these could indicate a reduced ability to perform usual activities.

## Future research

Our analyses indicated that symptoms cluster together in certain ways for certain treatment groups, but without providing an explanation for why symptoms within these clusters co-occurred. Some have suggested that certain symptoms may share an underlying mechanism [18, 20, 28] or one symptom that 'drives' them [19]. Future studies, building on and adding to the wider cluster research agenda [19] are thus warranted to enhance our understanding in this area, alongside qualitative studies to ascertain the clinical face validity of symptom clusters [17]. Ultimately, this should inform future studies into effective interventions that incorporate our suggested cluster-based approach to symptom assessments, complemented with clinical practice guidance on how to manage symptoms [12, 13, 46].

## Conclusion

This study identified symptom clusters related to lack of energy and mobility, gastrointestinal, skin, and mental health across CKD treatment groups, including both people on KRT and those with CKD but not on KRT. The 'lack of energy and mobility' clusters were consistently associated with having problems with usual activities across treatment groups. The study findings provide pointers for a more efficient cluster-based approach to symptom assessments by healthcare professionals. Future studies should focus on developing cluster-level symptom management interventions with enhanced potential to improve the care and outcomes for people with CKD.

## Supporting information

**S1 Table. STROBE statement [32]—checklist of items and where in the manuscript they have been included.**
(DOCX)

**S2 Table. Sociodemographic characteristics of people on kidney replacement treatments in the UK Renal Registry at 31st December 2016 for the participating centres.**
(DOCX)

**S3 Table. Baseline characteristics for people with ≥4 missing symptom or without a score for 'problems with usual activities' whose questionnaires were excluded from all analyses.**
(DOCX)

**S4 Table. Baseline characteristics for all people with CKD on KRT in the UK Renal Registry at 31st December 2016.**
(DOCX)

**S5 Table. Baseline characteristics of people with more than one survey.**
(DOCX)

**S6 Table. Principal component loadings of all symptoms prior to final cluster composition.**
(DOCX)

**S7 Table. Summary of changes total symptom cluster scores and changes in usual activities score and per treatment group.**
(DOCX)

## Acknowledgments

We would like to thank all people who completed the questionnaire and all kidney unit staff who facilitated the data collection.

## Author Contributions

**Conceptualization:** Currie Moore, Thomas J. Wilkinson, Fergus J. Caskey, Sabine N. van der Veer.

**Formal analysis:** Shalini Santhakumaran, Glen P. Martin, Winnie Magadi.

**Methodology:** Shalini Santhakumaran, Glen P. Martin, David Wellsted, Sabine N. van der Veer.

**Project administration:** Sabine N. van der Veer.

**Supervision:** Sabine N. van der Veer.

**Validation:** Fergus J. Caskey.

**Writing – original draft:** Currie Moore, Sabine N. van der Veer.

**Writing – review & editing:** Currie Moore, Shalini Santhakumaran, Glen P. Martin, Thomas J. Wilkinson, Fergus J. Caskey, Winnie Magadi, Rachel Gair, Alice C. Smith, David Wellsted, Sabine N. van der Veer.

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
