## [Decision Letter · Decision Letter 0]

10 Jun 2021

PONE-D-21-05457

Symptom clusters in chronic kidney disease and their association with patients’ ability to perform usual activities

PLOS ONE

Dear Dr. Moore,

Thank you for submitting your manuscript to PLOS ONE. After careful consideration, we feel that it has merit but does not fully meet PLOS ONE’s publication criteria as it currently stands. Therefore, we invite you to submit a revised version of the manuscript that addresses the points raised during the review process.

We look forward to receiving your revised manuscript.

Kind regards,

Boris Bikbov

Academic Editor

PLOS ONE

Journal Requirements:

Reviewers' comments:

Reviewer's Responses to Questions

**Comments to the Author**

1. Is the manuscript technically sound, and do the data support the conclusions?

Reviewer #1: Partly

Reviewer #2: Yes

Reviewer #3: Yes

2. Has the statistical analysis been performed appropriately and rigorously? 

Reviewer #1: I Don't Know

Reviewer #2: Yes

Reviewer #3: Yes

3. Have the authors made all data underlying the findings in their manuscript fully available?

Reviewer #1: Yes

Reviewer #2: Yes

Reviewer #3: No

4. Is the manuscript presented in an intelligible fashion and written in standard English?

Reviewer #1: Yes

Reviewer #2: Yes

Reviewer #3: Yes

5. Review Comments to the Author

Reviewer #1: Dear autor,

Several studies point to the worsening of quality of life in patients with chronic kidney disease and

I highlight some points of improvement in the manuscript:

1. The abstract, the methodology is not clear and objective and compromises the analysis of the results.

2. The background the introduction could focus more on the effects of symptoms and improve the quality of life of these patients.

3. Methods: This Topic could be more objective with a description of the applied methodology.

4. The statistical analyses: there was no robust statistical analysis.

5. this study uses data from 2017 and may be more outdated.

6. The author could discuss in more detail the follow-up of these patients and opportunities to apply and approach these symptoms.

7. Is important describe the sociodemographic data of these centers for explain the symptoms is regulary for everybody.

Thank you!

Reviewer #2: Dear Authors,

Thank you for an interesting manuscript and the following comments are provided to improve the manuscript.

Title, Abstract, Introduction and Background

- Please do not label patients by a disease or treatment (e.g. CKD patient groups, haemodialysis patient, TP-CKD participants, etc) – amend in the abstract and throughout the manuscript

- Please use the current international nomenclature for kidney disease published in the journal Kidney International (Levey et al 2020) – such as kidney replacement therapy.

- No need to use “our”, “we use” “we found”, “our findings” and so on – simply rephrase these types of sentences throughout the manuscript

- Add design and data collection methods (symptom tool, etc) into abstract

- Please write the abstract and the entire manuscript according to STROBE guidelines.

- Amend sentence (lines 81-83) as there is at least one study which included patients not receiving KRT (ref 20 - Almutary et al included almost 25% non-KRT [both CKD G4 and G5]).

- Almutary et al 2017 also constructed structural equation modelling demonstrating the direct (casual) relationships between symptom clusters and health-related quality of life

- Sentence (lines 85-86), the phrase “modest sample sizes” is vague – especially when cluster analysis is only possible when there is a sufficient sample size to do this type of statistical analysis. Arguably more studies on CKD symptom clusters because symptoms are subjective and could reflect particular cultures, health experiences, etc, etc. There is no need to avoid needing replication studies in another patient population to add to evidence.

- Clarify the aims related to longitudinal data collection.

Methods

- Ensure all STROBE aspects are explicitly described.

- For the TP-CKD program, at grade of CKD are people enrolled? If CKD G4 were included, then further clarity in data analysis and results will be required; that is, how were the different groups handled (i.e. CKD G4, CKD G5, KRT groups)

- Clarify how PROM data was collected.

- Clarify ethical procedures for TP-CKD.

- Data analysis – at what level was significance set?

Results

- Very thorough and interesting

Discussion

- First paragraph – there is no need to restate design, sample and sites for the study, instead interpret the meaning of the overall results (without restating actual numbers).

- Add to several paragraphs in the discussion what explanation can be made for the differences found between this study and previous studies on symptom clusters (why isn’t there more consistency which is only partly explained by different symptom measures used). C. Miaskowski research in cancer symptom clusters and H.A. de Von in CVD symptom clusters is likely to be insightful for this manuscript.

- Lack of energy and its similarity (or difference) to fatigue also requires further interpretation.

- In the discussion on this study’s symptoms and activity, it would also be worth reviewing and discussing with regard to Almutary et al 2017 CKD symptom cluster modelling which demonstrated significant casual relationships with physical function (and also with mental health)

Clinical Implications

- Clarify why the IPOS-Renal is not recommend as this PROM is widely used (rather than POS-S renal).

Future Research

- Again C. Miaskowski and colleagues are exploring underlying mechanisms in cancer symptoms and symptom cluster. There is an opportunity for researchers in CKD to draw from this work and extend it.

Reviewer #3: This was a study that used secondary analysis of data gathered as part of a national level research in the UK to examine association between symptom clusters and usual activities in chronic kidney disease (CKD).

The manuscript has been written according to the author guidelines. Introduction section provides a background and good overview of contemporary literature relevant to the study. Statistical analysis was adequately and appropriately described. Results, discussion and conclusions sections were described in line with study aims. The manuscript was well presented.

Minor comments

• Currently, the terminology of CKD has been revised, enabling researchers to compare results across studies conveniently. For example, end-stage kidney failure or end-stage kidney disease, now termed as kidney failure. Please refer...

Levey et al. (2020). Nomenclature for kidney function and disease: Executive summary and glossary from a Kidney Disease: Improving Global Outcomes (KDIGO) consensus conference. Journal of Nephrology, 33(4), 639–648. doi:10.1007/s40620-020-00773-6

Threfore, it is recommend to reword CKD terminology throughout the manuscript to fit current guidelines.

• It is also suggested not to label people living with chronic diseases as patients (e.g., line 89: CKD patient groups) consistently throughout the manuscript.

• Methods: Line 119: Spell out EQ5D5L first, before using it in abbreviated form.

• Methods: Line 128: It is not clear that what do you mean by ‘see below’ as there is no further information given in relation to self-rated health.

• Line numbers are missing after the line 266. Therefore, it is difficult to point comments on exact places.

• Under discussion, relation to other studies, second paragraph; it is better to mention frequently co-occurring mental health symptoms. i.e., anxiety and depression

• According to current evidence, fatigue is the most frequent symptom experienced by those with CKD. See for example, Almutary et al. (2013) and (2016) in your reference list. However, POS-S-renal doesn’t capture fatigue. It is note that weakness/lack of energy is different to fatigue. Therefore, it is important to discuss this point in the discussion under the section ‘relation to other studies’.

• Use the RRT abbreviation consistently throughout the manuscript. Abbreviated form of renal replacement therapy is missing in the conclusions section.

• It is suggested to check the reference list to align journal requirements.

Thank you for the opportunity given to review this manuscript.

6. PLOS authors have the option to publish the peer review history of their article (what does this mean?). If published, this will include your full peer review and any attached files.

Reviewer #1: No

Reviewer #2: No

Reviewer #3: **Yes: **Dr Harith Eranga Yapa

---

## [Author Response · Author response to Decision Letter 0]

18 Sep 2021

Point-by-point response to reviewers’ comments

Manuscript: Symptom clusters in chronic kidney disease and their association with patients’ ability to perform usual activities (PONE-D-21-05457)

Page and line numbers refer to those in the ‘clean’ version of the revised manuscript.

Reviewer #1

1. The abstract, the methodology is not clear and objective and compromises the analysis of the results.

We have revised the abstract and made several edits with the aim of making our study and methodology clearer.

2. The background the introduction could focus more on the effects of symptoms and improve the quality of life of these patients.

In the Introduction, we have clarified that symptoms, and particularly symptom clusters, are associated with poorer quality of life and health functioning and higher levels of depression, and that managing symptoms clusters may improve outcomes related to quality of life. For example, by including sentences such as:

• “Those with CKD may suffer from 6 to 20 symptoms simultaneously, which negatively affects quality of life and increases the risk of treatment non-adherence, health care utilization, and mortality.” (page 4, lines 63-65)

• “… previous research proposed clusters related to fatigue, pain, gastrointestinal, and skin symptoms. They also reported negative associations between these clusters and outcomes linked to quality of life, such as health functioning and depression. This implies that managing clusters, rather than individual symptoms, may be an effective way to improve outcomes related to quality of life for people with CKD…” (page 4, lines 75-80)

3. Methods: This Topic could be more objective with a description of the applied methodology.

We have further clarified our methodology by making several changes to the Methods section, such as clarifying:

• In the first sentence of the Methods that: “We conducted secondary analyses of cross-sectional and longitudinal data” (pg 5, lines 100-101)

• How patient-reported outcome data was collected: (pg 6, lines 116-117; pg 7, lines 143-144) “People who were interested received a paper copy of the questionnaire, which they returned to their unit upon completion. […] The completed TP-CKD paper-based questionnaires were sent to the UK Renal Registry (UKRR) and scanned into electronic format.”

• The assessment methods for demographic and treatment data (pg 7, lines 148-150): “The UKRR annual report describes the definitions and measurement methods for these data items [ref].” 

• Explaining why the self-reported health status question was not included in the analysis (pg 7, lines 136-141): “We did not consider the remaining EQ-5D-5L items as part of our the outcome measure […] because […] responses could not be scanned into electronic format (self-rated health status).”

• That we did not perform sensitivity analyses (pg 10, line 213): “We did not perform any sensitivity analyses.”

 

4. The statistical analyses: there was no robust statistical analysis.

We have described our approach to the statistical analyses in the Methods section under ‘Secondary data analysis’ (pg 8-10, lines 161-213). This approach is in keeping with previous studies that investigated symptom clusters in people with chronic kidney disease, such as Almutary et al 2016 and Thong et al 2009. We have referenced these studies in our manuscript. We have also clarified that “For all analyses, we used SAS version 9.4 for all analyses and considered a p value of <0.5 significant.” (pg 10, line 212-213).

5. This study uses data from 2017 and may be more outdated.

This is a valid point for which we have added the following sentence in the Limitations section of the Discussion to address this (page x, lines x-x):

“In addition, the data analysed in this study was collected in 2017, and we propose repeating the analysis in the future when updated registry data becomes available.” 

6. The author could discuss in more detail the follow-up of these patients and opportunities to apply and approach these symptoms.

In the Discussion, we have suggested to combine our proposed cluster-based approach to symptom assessment with existing clinical practice guidance on symptom management by adding the following sentence (pg 24, lines 361-363): 

“… effective interventions that incorporate our suggested cluster-based approach to symptom assessments, complemented with clinical practice guidance on how to manage symptoms [refs].”

7. Is important describe the sociodemographic data of these centers for explain the symptoms is regulary for everybody.

Thank you for this valuable suggestion. We have added a table presenting the sociodemographic characteristics for all people on a KRT treated by each of the participating centres (see Supplementary material, Table S1).

Reviewer #2

Title, Abstract, Introduction and Background

1. Please do not label patients by a disease or treatment (e.g. CKD patient groups, haemodialysis patient, TP-CKD participants, etc) – amend in the abstract and throughout the manuscript

We very much appreciate this advice and have addressed this throughout the manuscript as much as possible. For example, we changed ‘kidney patients’ to ‘people with CKD’, and we now state that we ‘categorised the data into treatment groups’, rather than ‘categorising participants into patient groups’.

2. Please use the current international nomenclature for kidney disease published in the journal Kidney International (Levey et al 2020) – such as kidney replacement therapy.

We very much appreciate the reviewer pointing us to Levey et al’s 2020 nomenclature. We have made considerable changes throughout the text and believe it is now in line with this nomenclature. We have also stated at the start of the Methods that we followed Levey et al for reporting our study findings (pg 5, lines 103-106).

 

3. No need to use “our”, “we use” “we found”, “our findings” and so on – simply rephrase these types of sentences throughout the manuscript

We reviewed the manuscript and identified all occasions where we could reasonably remove ‘our’, or replace ‘we’ by a noun without changing the meaning of sentences. Examples in the abstract include:

• “In our analysis, We categorised data by treatment group: …” (Methods section)

• Overall, we identified clusters related to: …” (Results section)

However, in most instances, we used ‘we’ to avoid passive voice, which is in line with all major style guides and advocated by most academic journals. We have therefore kept these phrases as they were.

4. Add design and data collection methods (symptom tool, etc) into abstract

Thanks for recommending this edit - we have now stated this in the abstract as follows: 

“We conducted a secondary analysis of both cross-sectional and longitudinal data collected as part of a national service improvement programme in 14 renal centres in England, UK. This data included symptom severity (17 items, POS-Renal) and the extent to which people had problems performing their usual activities (single item, EQ-5D-5L).”

5. Please write the abstract and the entire manuscript according to STROBE guidelines.

We have used the STROBE guideline to review our manuscript and revised it accordingly. For example, we have added information on:

• assessment methods for demographic and treatment data (STROBE item 8):“The UKRR annual report describes the definitions and measurement methods for these data items [ref].” 

• sensitivity analyses (STROBE item12c): “We did not perform any sensitivity analyses.”

• numbers of individuals at each stage of study (STROBE item 13). For example: “The UKRR received questionnaires from 3,421 people, of whom 282 were excluded because they had ≥4 missing symptom scores or did not complete the ‘usual activities’ item on EQ-5D-5L (see Supplementary material, Table S3 for their characteristics). The remaining 3,139 people who completed at least one TP-CKD questionnaire were included in the analysis.” (Results pg 10, lines 217-221). Where we did not have this information because of the nature of a secondary data analysis, we further clarified this in the Methods section. For example, “Centres did not record information on whom they screened for eligibility, who had been confirmed eligible but declined participation, or people’s reasons for declining participation.” (Methods pg 6, lines 117-119).

• the external validity of our findings (STROBE item 21) (Discussion pg 22, lines 306-309): “It provides a robust external validation of clusters identified by previous studies, including skin (21, 24, 26, 27), gastrointestinal (20, 22), and mental health (24, 26), which suggests that our findings are generalisable beyond the study context.”

Lastly, we have stated at the start of the Methods section that we used STROBE to guide the reporting of our study (pg 5, lines 103-106), and included a completed STROBE checklist in the Supplementary material (Table S1) to indicate on which page each STROBE item has been included.

6. Amend sentence (lines 81-83) as there is at least one study which included patients not receiving KRT (ref 20 - Almutary et al included almost 25% non-KRT [both CKD G4 and G5]).

We have added the 2016 Almutary et al reference to the Introduction and changed the sentence as follows to clarify that most previous studies identified symptom clusters across rather than stratified for treatments/stages (page 5, lines 83-91): 

“CKD symptom clusters have been mainly investigated in people receiving dialysis [refs]. Studies considering other kidney replacement therapies (KRTs) or CKD stages often identified clusters across rather than stratified by treatment or stage [refs incl 2016 Almutary]. … Only few studies looked at symptom clusters specifically for people with a kidney transplant or with CKD but not on KRT (CKD non-KRT)], despite these people reporting a similar symptom burden. This leaves it largely unknown to what extent current clusters generalise across or differ between treatment groups.”

7. Almutary et al 2017 also constructed structural equation modelling demonstrating the direct (casual) relationships between symptom clusters and health-related quality of life

We have added the 2017 Almutary et al study as a reference to the following sentence (pg 4, line 75-78): “In CKD, previous research proposed clusters related to fatigue, pain, gastrointestinal, and skin symptoms. They also reported negative associations between these clusters and patient outcomes linked to quality of life, such as health functioning [refs incl 2017 Almutary] and depression.”

8. Sentence (lines 85-86), the phrase “modest sample sizes” is vague – especially when cluster analysis is only possible when there is a sufficient sample size to do this type of statistical analysis. 

We agree that ‘modest sample sizes’ is vague and have further specified this by changing the sentence as follows (pg 5, lines 85-87): “…; one reason for this lack of stratification might have been the modest sample sizes of <450 people.”

9. Arguably more studies on CKD symptom clusters because symptoms are subjective and could reflect particular cultures, health experiences, etc, etc. There is no need to avoid needing replication studies in another patient population to add to evidence.

To better explain the rationale for our study, we have amended the following paragraph in the Introduction as follows (page 5, lines 83-91): 

“CKD symptom clusters have been mainly investigated in people receiving dialysis [refs]. Studies considering other kidney replacement therapies (KRTs) or CKD stages often identified clusters across rather than stratified by treatment or stage [refs]; one reason for this lack of stratification might have been the modest sample sizes of <450 people [refs]. Only few studies looked at symptom clusters specifically for people with a kidney transplant [refs] or with CKD but not on KRT (CKD non-KRT) [ref], despite them reporting a similar symptom burden [refs]. This leaves it largely unknown to what extent current clusters generalise across or differ between treatment groups.”

10. Clarify the aims related to longitudinal data collection.

Thank you for this helpful suggestion. We have clarified the aim of the longitudinal analyses in the the introduction as follows (pg 5, lines 92-95): 

“This study, therefore, aimed to (1) explore symptom clustering in a large data set, stratified by CKD treatment group,(2) assess the relevance of the identified clusters by investigating their association with people’s ability to perform their usual activities, and (3) determine if these associations were stable over time.”

Methods

11. Ensure all STROBE aspects are explicitly described.

See our response to reviewer 2 point 5

12. For the TP-CKD program, at grade of CKD are people enrolled? If CKD G4 were included, then further clarity in data analysis and results will be required; that is, how were the different groups handled (i.e. CKD G4, CKD G5, KRT groups)

The data set did not include information on CKD stage at enrollment in the non-KRT group. We have clarified this in footnote a) to Table 1 as follows:

“[the CKD non-KRT treatment group] included people with any stage of CKD not on KRT who were under the treatment of a kidney centre. The dataset did not contain information on the stage of CKD at enrollment”

13. Clarify how PROM data was collected.

We have clarified this by adding/changing the following sentences to the Methods (pg 6, lines 116-117; pg 7, lines 143-144): “People who were interested received a paper copy of the questionnaire, which they returned to their unit upon completion. [..] The completed TP-CKD paper-based questionnaires were sent to the UK Renal Registry (UKRR) and scanned into electronic format.”

14. Clarify ethical procedures for TP-CKD.

We have clarified that no formal ethical approval was required for TP-CKD (pg 7-8, lines 151-160): “Because the primary purpose of TP-CKD was service evaluation and improvement rather than research, no formal ethical approval was required. People taking part in the TP-CKD programme implicitly consented to their data being processed and linked by returning a completed questionnaire to their kidney centre. The UKRR holds permissions under s251 of the National Health Service (NHS) Act 2006, to gather, process, and share confidential patient information for the purposes of audit and research. These permissions are renewed annually by the UK’s Health Research Authority’s Confidentiality Advisory Group. The collection and secondary analysis of the data for this study were approved by the UKRR (ref: UKRR ILD32) and carried out under the ethical permissions granted to the UKRR by the Research Ethics Committee.”

15. Data analysis – at what level was significance set?

Thank you for noting this. We have added the following sentence to the end of the Methods section (pg 10, lines 212-213): “For all analyses, we used SAS version 9.4 for all analyses and considered a p value of <0.5 significant.”

Discussion

16. First paragraph – there is no need to restate design, sample and sites for the study, instead interpret the meaning of the overall results (without restating actual numbers).

We have removed the first sentence stating the design, sample and sites, and in line with STROBE, focused the first paragraph on a summary of findings for each of the study objectives (pg 22, lines 299-303): “This study found that, overall, CKD symptom clusters related to lack of energy and mobility, gastrointestinal, skin, and mental health. Although clusters varied between treatment groups, the ‘lack of energy and mobility’ clusters were consistently associated with having problems doing usual activities across groups, both cross-sectionally and longitudinally.”

17. Add to several paragraphs in the discussion what explanation can be made for the differences found between this study and previous studies on symptom clusters (why isn’t there more consistency which is only partly explained by different symptom measures used). C. Miaskowski research in cancer symptom clusters and H.A. de Von in CVD symptom clusters is likely to be insightful for this manuscript.

Thank you for pointing us to the highly relevant paper by Miaskowski et al. We have referenced this work in the Discussion’s ‘Relation to other studies’ section as follows:

• (pg 22, lines 305-311) “We used the largest data set to date to investigate symptom clusters across treatment groups in people with CKD. It provides a robust external validation of clusters across treatment groups identified by most previous studies, including skin, gastrointestinal, and mental health […]. This alignment with previous research on CKD symptom clusters is in contrast to other disease areas, such as cancer, where there seems to be less consistency in symptom clusters across studies [ref to Miaskowski].”

• (pg 22-23, lines 318-322): “Furthermore, the ‘uraemic’ cluster in other studies included nausea and poor appetite (20, 21, 26, 27), whereas in our study, these symptoms, together with vomiting, formed a gastrointestinal cluster. This finding strongly aligns with wider symptom cluster research commonly identifying a separate gastrointestinal cluster consisting of nausea and vomiting [ref to Miaskowski].”

We also provided an explanation for the differences between our findings and those of some others with regards to mental health as a separate cluster (pg 23, lines 333-339): 

“In the CKD non-KRT treatment group, we found that frequently co-occurring mental health symptoms, i.e. feeling anxious and feeling depressed, formed a stand-alone cluster, which differs from what Lee and Jeon reported in their smaller, Korean study. An explanation for mental health being a separate cluster may be that, as people approach kidney failure, they often experience more psychological symptoms. In the peritoneal dialysis group in the current study, this cluster also included pain and diarrhoea, both of which may negatively affect mental health.”

18. Lack of energy and its similarity (or difference) to fatigue also requires further interpretation.

Thank you for this constructive suggestion. We have added the following section to the Discussion to address this (pg 23, lines 325-332): 

“The POS-S Renal also does not include fatigue, which is a common symptom in people with CKD that others found to pertain to several symptom clusters. Lack of energy and weakness represent some but not all aspects of fatigue, which is a multi-faceted symptom related to sociodemographic, psychological, clinical and biochemical factors. This implies that part of the people reporting lack of energy and weakness in our study might have actually suffered from fatigue, which may partly explain the strong associations we found between ‘lack of energy and weakness’ clusters and people’s ability to do usual activities.”

19. In the discussion on this study’s symptoms and activity, it would also be worth reviewing and discussing with regard to Almutary et al 2017 CKD symptom cluster modelling which demonstrated significant casual relationships with physical function (and also with mental health)

We agree this is a relevant study to reference in this context. We have added it to the following statement in the Discussion (pg 23, line 340-343): 

“Our analyses showed that the ‘lack of energy and mobility’ clusters were associated with problems performing usual activities in all groups and models, which echoes the findings of previous research in which uraemic clusters were associated with impaired physical and mental functioning [refs incl Almutary et al 2017].”

Clinical Implications

20. Clarify why the IPOS-Renal is not recommend as this PROM is widely used (rather than POS-S renal).

The two main reasons for using the POS-S Renal instead of the IPOS-Renal are:

1. The POS-S Renal contains 17 common CKD symptoms to use for composing clusters, while the IPOS-Renal only includes a subset of these symptoms. 

2. The IPOS-Renal includes additional items on concerns beyond symptoms, such as information needs and practical issues. These items would not have been included in the symptom cluster analysis. 

We propose not to include this explanation in the manuscript because there is a risk it may cause confusion rather than provide clarity.

Future Research

21. C. Miaskowski and colleagues are exploring underlying mechanisms in cancer symptoms and symptom cluster. There is an opportunity for researchers in CKD to draw from this work and extend it.

Thank you for this helpful suggestion. We have added a more explicit reference to Miaskowski’s paper in the Discussion as follows (pg 27, lines 419-422):

“Some have suggested that certain symptoms may share an underlying mechanism or one symptom that ‘drives’ them. Future studies, building on and adding to the wider cluster research agenda [ref to Miaskowski], are warranted to enhance our understanding in this area, alongside qualitative studies to ascertain the clinical face validity of symptom clusters.”

Reviewer #3

1. Currently, the terminology of CKD has been revised, enabling researchers to compare results across studies conveniently. For example, end-stage kidney failure or end-stage kidney disease, now termed as kidney failure. Please refer. Levey et al. (2020). Therefore, it is recommend to reword CKD terminology throughout the manuscript to fit current guidelines.

We very much appreciate this advice and pointing us to the Levey et al 2020 nomenclature. We have made considerable changes throughout the text and believe it is now in line with this nomenclature. We have also stated at the start of the Methods that we followed Levey et al for reporting our study findings (pg 5, lines 103-106).

2. It is also suggested not to label people living with chronic diseases as patients (e.g., line 89: CKD patient groups) consistently throughout the manuscript.

We very much appreciate this advice and have addressed this throughout the manuscript as much as possible. For example, we changed ‘kidney patients’ to ‘people with CKD’, and we now state that we ‘categorised the data into treatment groups’, rather than ‘categorising participants into patient groups’.

3. Methods: Line 119: Spell out EQ5D5L first, before using it in abbreviated form.

Thank you for spotting this – we have now spelled it out on first mention as follows (pg 7, line 131-132):

“As the outcome measure we used an item from the EuroQol 5 dimensions - 5 level (EQ-5D-5L) version, which was also included in the TP-CKD questionnaire.”

 

4. Methods: Line 128: It is not clear that what do you mean by ‘see below’ as there is no further information given in relation to self-rated health.

We agree that this was not very clear in the text. We have removed the ‘see below’ reference and instead explained why the self-reported health status questions was not included in the analysis (pg 7, lines 136-141): 

“We did not consider the remaining EQ-5D-5L items as part of our the outcome measure [..] because [… responses could not be scanned into electronic format (self-rated health status).”

5. Under discussion, relation to other studies, second paragraph; it is better to mention frequently co-occurring mental health symptoms. i.e., anxiety and depression

We have taken your suggestion on board and reworded this sentence as follows (pg 23, lines333-334): 

“In the CKD non-KRT treatment group, we found that frequently co-occurring mental health symptoms, i.e. feeling anxious and feeling depressed, formed a stand-alone cluster, …”

6. According to current evidence, fatigue is the most frequent symptom experienced by those with CKD. See for example, Almutary et al. (2013) and (2016) in your reference list. However, POS-S-renal doesn’t capture fatigue. It is note that weakness/lack of energy is different to fatigue. Therefore, it is important to discuss this point in the discussion under the section ‘relation to other studies’.

Thank you for this constructive suggestion. We have added the following sentences to the Discussion Section ‘Relation to Other Studies’ to address this (pg 23, lines 325-332): 

“The POS-S Renal also does not include fatigue, which is a common symptom in people with CKD that others found to pertain to several symptom clusters. Lack of energy and weakness represent some but not all aspects of fatigue, which is a multi-faceted symptom related to sociodemographic, psychological, clinical and biochemical factors. This implies that part of the people reporting lack of energy and weakness in our study might have actually suffered from fatigue, which may partly explain the strong associations we found between ‘lack of energy and weakness’ clusters and people’s ability to do usual activities.”

7. Use the RRT abbreviation consistently throughout the manuscript. Abbreviated form of renal replacement therapy is missing in the conclusions section.

Thank you for spotting this. We have changed renal replacement therapy to KRT in line with rest of the manuscript.

8. It is suggested to check the reference list to align journal requirements.

We have corrected instances where references were not in line with journal requirements.

---

## [Decision Letter · Decision Letter 1]

14 Oct 2021

PONE-D-21-05457R1Symptom clusters in chronic kidney disease and their association with people’s ability to perform usual activitiesPLOS ONE

Dear Dr. Moore,

Thank you for submitting your manuscript to PLOS ONE. After careful consideration, we feel that it has merit but does not fully meet PLOS ONE’s publication criteria as it currently stands. Therefore, we invite you to submit a revised version of the manuscript that addresses the points raised during the review process.

 Please submit your revised manuscript by Nov 28 2021 11:59PM. If you will need more time than this to complete your revisions, please reply to this message or contact the journal office at plosone@plos.org. Please include the following items when submitting your revised manuscript:A rebuttal letter that responds to each point raised by the academic editor and reviewer(s). You should upload this letter as a separate file labeled 'Response to Reviewers'.A marked-up copy of your manuscript that highlights changes made to the original version. You should upload this as a separate file labeled 'Revised Manuscript with Track Changes'.An unmarked version of your revised paper without tracked changes. You should upload this as a separate file labeled 'Manuscript'.If applicable, we recommend that you deposit your laboratory protocols in protocols.io to enhance the reproducibility of your results. Protocols.io assigns your protocol its own identifier (DOI) so that it can be cited independently in the future. For instructions see: https://journals.plos.org/plosone/s/submission-guidelines#loc-laboratory-protocols. Additionally, PLOS ONE offers an option for publishing peer-reviewed Lab Protocol articles, which describe protocols hosted on protocols.io. Read more information on sharing protocols at https://plos.org/protocols?utm_medium=editorial-email&utm_source=authorletters&utm_campaign=protocols.

We look forward to receiving your revised manuscript.

Kind regards,

Boris Bikbov

Academic Editor

PLOS ONE

Journal Requirements:

Reviewers' comments:

Reviewer's Responses to Questions

**Comments to the Author**

1. If the authors have adequately addressed your comments raised in a previous round of review and you feel that this manuscript is now acceptable for publication, you may indicate that here to bypass the “Comments to the Author” section, enter your conflict of interest statement in the “Confidential to Editor” section, and submit your "Accept" recommendation.

Reviewer #1: All comments have been addressed

Reviewer #2: All comments have been addressed

Reviewer #3: All comments have been addressed

2. Is the manuscript technically sound, and do the data support the conclusions?

Reviewer #1: Partly

Reviewer #2: Yes

Reviewer #3: Yes

3. Has the statistical analysis been performed appropriately and rigorously? 

Reviewer #1: Yes

Reviewer #2: Yes

Reviewer #3: Yes

4. Have the authors made all data underlying the findings in their manuscript fully available?

Reviewer #1: (No Response)

Reviewer #2: No

Reviewer #3: Yes

5. Is the manuscript presented in an intelligible fashion and written in standard English?

Reviewer #1: Yes

Reviewer #2: Yes

Reviewer #3: Yes

6. Review Comments to the Author

Reviewer #1: Dear author, I recommend reviewing the abstract needs to include that I work with adults.

For methods you need We have clarified that no formal ethical approval was required for TP-CKD and the follow-up period.

Do you need to include more information about the inclusion and exclusion criteria (age, therapy, literate, other condition).

Describe in the methodology how the questionnaires were applied.

13. Clarify how PROM data was collected, this topic needs to be clear.

this topic is still unclear: 10. Clarify the aims related to longitudinal data collection.

In the discussion, check whether the treatment time also interferes with symptoms.

Reviewer #2: Dear Authors,

Thank you for addressing all of my previous comments and suggestions. The manuscript is much improved and will make an additional contribution to the emerging science on CKD symptom clusters.

Reviewer #3: This was a study that used secondary analysis of data gathered as part of a national level research in the UK to examine association between symptom clusters and usual activities in chronic kidney disease. The manuscript has been written according to the author guidelines. Authors substantially improved the manuscript based on comments provided in the previous submission. The manuscript is well presented. Great work!

7. PLOS authors have the option to publish the peer review history of their article (what does this mean?). If published, this will include your full peer review and any attached files.

Reviewer #1: No

Reviewer #2: **Yes: **Ann Bonner

Reviewer #3: **Yes: **Dr Harith Eranga Yapa

---

## [Author Response · Author response to Decision Letter 1]

22 Oct 2021

Point-by-point response to reviewers’ comments

Manuscript: Symptom clusters in chronic kidney disease and their association with patients’ ability to perform usual activities (PONE-D-21-05457)

Page and line numbers refer to those in the ‘clean’ version of the revised manuscript.

Reviewer #1: 

Dear author, I recommend reviewing the abstract needs to include that I work with adults. 

Thank you for your suggestion. We have made it clear in the abstract (Line 29) that the data presented in the manuscript is with adults. In the Methods (Line 118), it is stated that people were eligible if they were ‘18 years or over’. 

For methods you need We have clarified that no formal ethical approval was required for TP-CKD and the follow-up period. 

In our previous Response to Reviewers, we revised the ethics section of the Methods to directly address this (see below - Lines 151-160). No formal ethical approval is required in the UK for service evaluation and improvement, regardless if it carried out in a cross-sectionally or longitudinally (like the TP-CKD).

We have revised Line 154 to indicate that no formal ethical approval was required if people completed more than one questionnaire:

“Because the primary purpose of TP-CKD was service evaluation and improvement rather than research, no formal ethical approval was required. People taking part in the TP-CKD programme implicitly consented to their data being processed and linked by returning a completed questionnaire(s) to their kidney centre.” Lines 151-154

Do you need to include more information about the inclusion and exclusion criteria (age, therapy, literate, other condition). 

The full inclusion and exclusion criteria are listed in Lines 112-113: People were eligible if they were aged 18 years or over and receiving care for any stage of CKD or on any form of KRT.

To confirm, there were no other inclusion and exclusion criteria for TP-CKD.

Describe in the methodology how the questionnaires were applied. 

Key methods for the questionnaires are described in lines 100-141 of the manuscript:

We conducted a secondary analyses of cross-sectional and longitudinal data collected in the context of a national service improvement programme in 14 kidney centres across England (UK) called Transforming Participation in Chronic Kidney Disease (TP-CKD) [31]. We followed the Strengthening the Reporting of Observational Studies in Epidemiology (STROBE) guidelines [32] (see Supplementary material, Table S1) and the nomenclature for kidney function and disease proposed by Levey and colleagues [33] for reporting our findings. 

Transforming Participation in Chronic Kidney Disease (TP-CKD)

The TP-CKD programme aimed to support people with CKD to better manage and make decisions about their own care and treatment. This included introducing collection of patient-reported outcome data in 14 English kidney centres [34]. Between December 2015 and December 2017, members of local kidney care teams approached eligible patients in their centre. People were eligible if they were aged 18 years or over and receiving care for any stage of CKD or on any form of KRT. The number and sociodemographic characteristics of people on KRT treated in each centre are provided in Supplementary material, Table S2; this does not include people with CKD not on KRT because this data was not available. People who were interested received a paper copy of the questionnaire, which they returned to their unit upon completion. Centres did not record information on whom they screened for eligibility, who had been confirmed eligible but declined participation, or on reasons for declining participation. In twelve centres, people who had previously taken part were invited to complete the questionnaire again at a later time.

Measures of exposure and outcome

In the current secondary analysis, we used symptom burden as the measure of exposure. To assess exposure, we analysed data collected through the Palliative care Outcome Scale-Symptom (POS-S) Renal [35], which was one of the patient-reported outcome measures included in the paper-based TP-CKD questionnaire. The POS-S Renal consists of 17 symptoms that are common for people with CKD [35], such as pain, weakness or lack of energy, and itching. For each symptom, respondents are asked to indicate to what extent they have been bothered by it over the last week on a scale from 0 (not at all) to 4 (overwhelmingly). 

As the outcome measure we used an item from the EuroQol 5 dimensions - 5 level (EQ-5D-5L) version [36], which was also included in the TP-CKD questionnaire. The EQ-5D-5L assesses general health status and asks respondents to rate whether they had problems doing their usual activities (e.g. work, study, housework, leisure activities) on a scale from 1 (no problem) to 5 (unable). People with CKD previously reported this item as being important to ‘living well’ (29). We did not consider the remaining EQ-5D-5L items as part of the outcome measure because: some fully overlapped with symptoms in the POS-S Renal (pain/discomfort; anxiety/depression; mobility); one was not reported by people with CKD as a priority outcome in research (self-care: wash and dress) [37,38]; and one because its responses could not be scanned into electronic format (self-rated health status).

13. Clarify how PROM data was collected, this topic needs to be clear.

this topic is still unclear: 

PROM data were collected as part of the TP-CKD survey which is described in lines 100-141 of the manuscript:

We conducted a secondary analyses of cross-sectional and longitudinal data collected in the context of a national service improvement programme in 14 kidney centres across England (UK) called Transforming Participation in Chronic Kidney Disease (TP-CKD) [31]. We followed the Strengthening the Reporting of Observational Studies in Epidemiology (STROBE) guidelines [32] (see Supplementary material, Table S1) and the nomenclature for kidney function and disease proposed by Levey and colleagues [33] for reporting our findings. 

Transforming Participation in Chronic Kidney Disease (TP-CKD)

The TP-CKD programme aimed to support people with CKD to better manage and make decisions about their own care and treatment. This included introducing collection of patient-reported outcome data in 14 English kidney centres [34]. Between December 2015 and December 2017, members of local kidney care teams approached eligible patients in their centre. People were eligible if they were aged 18 years or over and receiving care for any stage of CKD or on any form of KRT. The number and sociodemographic characteristics of people on KRT treated in each centre are provided in Supplementary material, Table S2; this does not include people with CKD not on KRT because this data was not available. People who were interested received a paper copy of the questionnaire, which they returned to their unit upon completion. Centres did not record information on whom they screened for eligibility, who had been confirmed eligible but declined participation, or on reasons for declining participation. In twelve centres, people who had previously taken part were invited to complete the questionnaire again at a later time.

Measures of exposure and outcome

In the current secondary analysis, we used symptom burden as the measure of exposure. To assess exposure, we analysed data collected through the Palliative care Outcome Scale-Symptom (POS-S) Renal [35], which was one of the patient-reported outcome measures included in the paper-based TP-CKD questionnaire. The POS-S Renal consists of 17 symptoms that are common for people with CKD [35], such as pain, weakness or lack of energy, and itching. For each symptom, respondents are asked to indicate to what extent they have been bothered by it over the last week on a scale from 0 (not at all) to 4 (overwhelmingly). 

As the outcome measure we used an item from the EuroQol 5 dimensions - 5 level (EQ-5D-5L) version [36], which was also included in the TP-CKD questionnaire. The EQ-5D-5L assesses general health status and asks respondents to rate whether they had problems doing their usual activities (e.g. work, study, housework, leisure activities) on a scale from 1 (no problem) to 5 (unable). People with CKD previously reported this item as being important to ‘living well’ (29). We did not consider the remaining EQ-5D-5L items as part of the outcome measure because: some fully overlapped with symptoms in the POS-S Renal (pain/discomfort; anxiety/depression; mobility); one was not reported by people with CKD as a priority outcome in research (self-care: wash and dress) [37,38]; and one because its responses could not be scanned into electronic format (self-rated health status).

10. Clarify the aims related to longitudinal data collection. 

We addressed this by explicitly stating the aims of the longitudinal analysis in lines 92-97 of the Introduction:

“This study, therefore, aimed to (1) explore symptom clustering in a large data set,

stratified by CKD treatment group,(2) assess the relevance of the identified clusters by

investigating their association with people’s ability to perform their usual activities, and

(3) determine if these associations were stable over time.” Lines 92-97

In the discussion, check whether the treatment time also interferes with symptoms.

In this study we aimed to explore symptom clusters and the association between these clusters and the ability to perform usual activities. We described time on KRT for the participants, and adjusted for time on KRT when exploring the association between symptoms and ability to perform usual activities. The effect of time on KRT on symptom burden was not an aim of this work so we have not included it in the discussion.

Reviewer #2: 

No further comments to address.

Thank you, Dr. Bonner, for your constructive feedback on the manuscript.

Reviewer #3: 

No further comments to address.

We appreciate your constructive feedback and kind words on the manuscript, Dr. Yapa.

---

## [Decision Letter · Decision Letter 2]

22 Nov 2021

PONE-D-21-05457R2Symptom clusters in chronic kidney disease and their association with people’s ability to perform usual activitiesPLOS ONE

Dear Dr. Moore,

Thank you for submitting your manuscript to PLOS ONE. After careful consideration, we feel that it has merit but does not fully meet PLOS ONE’s publication criteria as it currently stands. Therefore, we invite you to submit a revised version of the manuscript that addresses the points raised during the review process.

We look forward to receiving your revised manuscript.

Kind regards,

Boris Bikbov

Academic Editor

PLOS ONE

Journal Requirements:

Reviewers' comments:

Reviewer's Responses to Questions

**Comments to the Author**

1. If the authors have adequately addressed your comments raised in a previous round of review and you feel that this manuscript is now acceptable for publication, you may indicate that here to bypass the “Comments to the Author” section, enter your conflict of interest statement in the “Confidential to Editor” section, and submit your "Accept" recommendation.

Reviewer #1: All comments have been addressed

2. Is the manuscript technically sound, and do the data support the conclusions?

Reviewer #1: Partly

3. Has the statistical analysis been performed appropriately and rigorously? 

Reviewer #1: I Don't Know

4. Have the authors made all data underlying the findings in their manuscript fully available?

Reviewer #1: Yes

5. Is the manuscript presented in an intelligible fashion and written in standard English?

Reviewer #1: Yes

6. Review Comments to the Author

Reviewer #1: This manuscript is a cross-sectional analysis of patients undergoing renal treatment.

In the Transforming Participation in Chronic Kidney Disease (TP-CKD) session, better describe the program, how were the patients asked to fill out and is there an interval for re-application of the questionnaires?

Such missing information compromises the analysis of table 3. Another important data for table 3 is to include the time between analyses. Several studies point to the impact of treatment time and outcomes of interest.

Data on frequency, initiation of this analysis and inclusion of this questionnaire in renal care services are not described.

Is there a protocol for applying the questionnaire annually?

lines 117 to 121:

Interested people received a printed copy of the questionnaire, which they returned to the unit after completing it. Sites did not record information about who they screened for eligibility, who was confirmed eligible but refused participation, or about the reasons for denial. In twelve centers, people who had already participated were invited to answer the questionnaire later.

It is important to inform the interval between responses for the analysis of table 3. The symptoms improved, which may be a reflection of the treatment and others not, such finding may be due to the treatment process itself.

Such observations should be placed as limitations, as well as the non-homogeneous adherence of the centers.

In the discussion, the main symptom identified 'lack of energy and mobility' may be related to the treatment time and its impact, which must be considered, long therapies, displacements...

7. PLOS authors have the option to publish the peer review history of their article (what does this mean?). If published, this will include your full peer review and any attached files.

Reviewer #1: No

---

## [Author Response · Author response to Decision Letter 2]

24 Dec 2021

Point-by-point response to reviewers’ comments

Manuscript: Symptom clusters in chronic kidney disease and their association with patients’ ability to perform usual activities (PONE-D-21-05457R2)

Note: Page and line numbers refer to those in the ‘clean’ version of the manuscript.

Reviewer #1

1. This manuscript is a cross-sectional analysis of patients undergoing renal treatment. In the Transforming Participation in Chronic Kidney Disease (TP-CKD) session, better describe the program, how were the patients asked to fill out and is there an interval for re-application of the questionnaires? Such missing information compromises the analysis of table 3.

In response to reviewer comments on previous versions of the manuscript, we endeavoured to provide the necessary background information on the TP-CKD programme in a dedicated section in the Methods (Ln 107-121): 

“The TP-CKD programme aimed to support people with CKD to better manage and make decisions about their own care and treatment. This included introducing collection of patient-reported outcome data in 14 English kidney centres [34]. Between December 2015 and December 2017, members of local kidney care teams approached eligible patients in their centre. People were eligible if they were aged 18 years or over and receiving care for any stage of CKD or on any form of KRT. The number and sociodemographic characteristics of people on KRT treated in each centre are provided in Supplementary material, Table S2; this does not include people with CKD not on KRT because this data was not available. People who were interested received a paper copy of the questionnaire, which they returned to their unit upon completion. Centres did not record information on whom they screened for eligibility, who had been confirmed eligible but declined participation, or on reasons for declining participation. In twelve centres, people who had previously taken part were invited to complete the questionnaire again at a later time.”

2. Another important data for table 3 is to include the time between analyses. Several studies point to the impact of treatment time and outcomes of interest.

Table 3 (p. 20) presents the results of the cross-sectional analysis of the data, and as such, there is no time element for this part of our analyses. For the longitudinal analysis (presented in Table 4, p. 21), we reported the time between measurements in the text (Ln 234-5): “A subsample of 699 people completed follow-up questionnaires at a median of 203 days (IQR, 133 to 301) after baseline, while remaining in the same treatment group.”

3. Data on frequency, initiation of this analysis and inclusion of this questionnaire in renal care services are not described. Is there a protocol for applying the questionnaire annually?

In the Methods section (Ln 107-121; also see response to point 1), we provided details of the TP-CKD programme with regard to when the data was collected, in whom and how. As described in the Methods, the programme ran between December 2015 and December 2017 (Ln 110-111). We this we implied there is no protocol for applying the questionnaire annually.

4. lines 117 to 121: “Interested people received a printed copy of the questionnaire, which they returned to the unit after completing it. Sites did not record information about who they screened for eligibility, who was confirmed eligible but refused participation, or about the reasons for denial. In twelve centers, people who had already participated were invited to answer the questionnaire later.” It is important to inform the interval between responses for the analysis of table 3. The symptoms improved, which may be a reflection of the treatment and others not, such finding may be due to the treatment process itself. Such observations should be placed as limitations, as well as the non-homogeneous adherence of the centers.

Table 3 (p. 20) presents the results of the cross-sectional analysis of the data, and as such, there is no time element for this part of our analyses. For the longitudinal analysis (presented in Table 4, p. 21), we reported the time between measurements in the text (Ln 234-5): “A subsample of 699 people completed follow-up questionnaires at a median of 203 days (IQR, 133 to 301) after baseline, while remaining in the same treatment group.”

In relation to the symptoms having improved, we would like to clarify that the aim of our study was to identify symptom clusters and their relationship to people’s performing daily activities, both cross-sectionally and longitudinally. The aim was not to describe symptom trajectories over time and as such, we do not report whether symptoms improved in the main text of the manuscript. As background information to the longitudinal analysis, we did include information in the supplement (Table S7) on how symptoms and people’s ability to perform daily activities changed over time. However, this information suggests that symptoms only improved over time for approximately half of patients, while they deteriorated for the other half. We, therefore, do not think these observations indicate clear limitations or ‘non-homogeneous adherence of centres’.

5. In the discussion, the main symptom identified 'lack of energy and mobility' may be related to the treatment time and its impact, which must be considered, long therapies, displacements...

To address the impact of treatment time, we have included ‘time on kidney replacement therapy’ in the partially and fully adjusted models of all analyses (see Table 3 (p. 20) and Table 4 (p. 21) to account for the influence this factor can have on symptoms.

---

## [Editor Report · Decision Letter 3]

9 Feb 2022

Symptom clusters in chronic kidney disease and their association with people’s ability to perform usual activities

PONE-D-21-05457R3

Dear Dr. Moore,

We’re pleased to inform you that your manuscript has been judged scientifically suitable for publication and will be formally accepted for publication once it meets all outstanding technical requirements.

Kind regards,

Boris Bikbov

Academic Editor

PLOS ONE

---

## [Editor Report · Acceptance letter]

18 Feb 2022

PONE-D-21-05457R3 

Symptom clusters in chronic kidney disease and their association with people’s ability to perform usual activities 

Dear Dr. Moore:

I'm pleased to inform you that your manuscript has been deemed suitable for publication in PLOS ONE. Congratulations! Your manuscript is now with our production department. 

Kind regards, 

on behalf of

Dr. Boris Bikbov 

Academic Editor

PLOS ONE